# Nicotine aversion is mediated by GABAergic interpeduncular nucleus inputs to laterodorsal tegmentum

Shannon L. Wolfman[1], Daniel F. Gill[1], Fili Bogdanic[2], Katie Long[3], Ream Al-Hasani[4], Jordan G. McCall[4,5,6], Michael R. Bruchas[5,6] & Daniel S. McGehee[1,2]

Nicotine use can lead to dependence through complex processes that are regulated by both its rewarding and aversive effects. Recent studies show that aversive nicotine doses activate excitatory inputs to the interpeduncular nucleus (IPN) from the medial habenula (MHb), but the downstream targets of the IPN that mediate aversion are unknown. Here we show that IPN projections to the laterodorsal tegmentum (LDTg) are GABAergic using optogenetics in tissue slices from mouse brain. Selective stimulation of these IPN axon terminals in LDTg in vivo elicits avoidance behavior, suggesting that these projections contribute to aversion. Nicotine modulates these synapses in a concentration-dependent manner, with strong enhancement only seen at higher concentrations that elicit aversive responses in behavioral tests. Optogenetic inhibition of the IPN–LDTg connection blocks nicotine conditioned place aversion, suggesting that the IPN–LDTg connection is a critical part of the circuitry that mediates the aversive effects of nicotine.

[1] Committee on Neurobiology, University of Chicago, Chicago, IL 60637, USA. [2] Department of Anesthesia & Critical Care, University of Chicago, Chicago, IL 60637, USA. [3] Interdisciplinary Scientist Training Program, University of Chicago, Chicago, IL 60637, USA. [4] St. Louis College of Pharmacy, Center for Clinical Pharmacology and Division of Basic Research of the Department of Anesthesiology, Washington University School of Medicine, St. Louis, MO 63110, USA. [5] Division of Basic Research, Department of Anesthesiology, Washington University Pain Center, St. Louis, MO 63110, USA. [6] Department of Neuroscience, Washington University School of Medicine, St. Louis, MO 63110, USA. Correspondence and requests for materials should be addressed to D.S.M.(email: dmcgehee@uchicago.edu)

Nicotine addiction remains a major public health problem worldwide, and treatments for smokers who want to quit remain only marginally effective[1]. The reinforcing effects of nicotine are well documented in both humans and laboratory animals[2], but at higher doses, nicotine also has intensely aversive effects[2,3]. In fact, humans and laboratory animals regulate nicotine intake levels when self-administering[4,5] and will maintain behaviors that prevent the administration of high nicotine doses[2]. Initial responses to nicotine may impact future dependence. A pleasurable first experience correlates with heavier smoking and higher rates of nicotine dependence than does an aversive first experience[6–10]. Similarly, a less aversive first experience may promote further use, supporting the transition from use to abuse and dependence[6,11]. Therefore, the balance between these initial rewarding and aversive effects may modulate subsequent nicotine use, leading to the development and maintenance of nicotine addiction[6,8,9,12].

The reinforcing effects of nicotine are mediated by high-affinity nicotinic acetylcholine receptors (nAChRs) expressed in several brain areas, most notably the mesoaccumbens dopamine system[13]. Within that circuitry, nAChR-induced increases in neuronal excitability and synaptic plasticity are critical for the reinforcing effects of the drug[14–17]. At higher nicotine concentrations, aversive effects are elicited, and recent studies suggest that lower-affinity nAChR subtypes expressed in the medial habenula (MHb) and its target the interpeduncular nucleus (IPN) stimulate distinct cellular mechanisms and pathways mediating avoidance behaviors[18–22]. In fact, the discovery of genetic variants in the CHRNA5-CHRNA3-CHRNB4 gene cluster associated with heavy smoking and higher relapse risk in humans[23–25] has led to enhanced focus on the MHb and IPN as a critical relay in the control of nicotine dependence[18,22,26–29].

Both the MHb and IPN densely express many of the known nAChR subunit genes[26,30], and several important studies have shown that increased activation of the MHb or IPN mediates aversion to high nicotine doses[18,22,28,30]. For example, knocking out the α5 nAChR subunit in mice, which limits the effects of high nicotine doses on MHb–IPN activity, promotes self-administration of high nicotine doses that wild-type mice do not administer, and this difference was eliminated by α5 rescue specifically in the MHb[22]. α5 knockout mice also develop a conditioned place preference to high doses of nicotine that wild-type mice do not, and they do not develop conditioned place aversion (CPA) to high doses of nicotine that wild-type mice do[31,32]. Decreasing activation of IPN neurons by knocking out the glucagon-like peptide-1 (GLP-1) receptor also results in self-administration of higher nicotine doses than wild-type mice administer[28].

In line with these findings, high, aversive doses of nicotine activate the IPN while lower doses fail to do so[22], and enhancing nicotine-induced excitation of MHb neurons by overexpressing β4 subunits of the putative low-affinity nAChRs results in CPA to a dose of nicotine that is neutral in control animals[18]. These studies suggest the expression of low-affinity nAChRs in the MHb and IPN that are not significantly activated by rewarding nicotine doses. Thus the MHb–IPN connection may dictate the upper limits of nicotine intake, likely via signals to other brain regions. Although recent anatomical investigations have clarified key IPN projections[33–35], the relevant downstream targets of the IPN and the mechanisms underlying aversion signaling remain unclear.

Here we explore the role of synaptic connections from the IPN to the laterodorsal tegmentum (LDTg) in nicotine aversion. The LDTg sends strong excitatory inputs to the ventral tegmental area (VTA) dopamine system, and these projections have been implicated in dopamine neuron firing rates, reward-related behavior, and addiction-related synaptic plasticity[36–39]. We find that optogenetic stimulation of the synaptic connections from the IPN to the LDTg are GABAergic and that selective activation of these synapses in vivo elicits avoidance behavior. These IPN–LDTg synaptic connections are enhanced by high concentrations of nicotine, but not by lower, rewarding concentrations. Optogenetic inhibition of the IPN–LDTg connection blocks CPA to a high dose of nicotine. Together, our findings suggest that the IPN–LDTg connection is a critical part of the circuitry that mediates the aversive effects of nicotine.

## Results

**Activation of IPN neurons results in behavioral avoidance.** Although there is evidence that IPN activation elicits aversion[18,22,40], some contradictions exist[41,42]. To test this, we expressed channelrhodopsin (ChR2) in the IPN, using a pan-neuronal promoter. ChR2 expression was confirmed visually and functionally with electrophysiology (Fig. 1a, b, Supplementary Fig. 1). We then placed fiber optic implants directly above the IPN (Fig. 1c, Supplementary Fig. 2A) and optogenetically stimulated IPN neurons during a real-time preference test (RTPT) (Fig. 1). Strong (20 Hz) IPN stimulation resulted in significant behavioral avoidance (Fig. 1d–g), while weak (1 Hz) stimulation did not, suggesting that strong, cell-type-independent activation of IPN neurons en masse results in aversion.

**Functional IPN inputs to LDTg are GABAergic.** To determine which IPN projection target might be relevant for aversion, we characterized the connection between the IPN and the LDTg[30] using brain slice electrophysiology. The LDTg performs many functions, including promoting reward via excitatory projections to VTA dopamine neurons[37]. Heterogeneous neurotransmission from IPN to LDTg has been reported, but the nature of this connection has not yet been functionally assessed[33].

We expressed ChR2 non-specifically in IPN neurons and waited 6 weeks to ensure terminal expression of the protein. We recorded from LDTg neurons that were retrogradely labeled from the VTA using fluorescent microbeads (Fig. 2a, b, Supplementary Figs. 1, 3A-C). Photo-stimulation of IPN terminals in the LDTg (Supplementary Fig. 3D) evoked synaptic currents that were blocked by the GABA$_A$ antagonist bicuculline (20 μM) and had reversal potentials close to $E_{Cl}$ (Fig. 2b–d). Therefore, IPN inputs to VTA-projecting LDTg neurons are GABAergic.

**IPN inputs to LDTg mediate aversion.** To test the role of this connection in aversion, we again used the RTPT, this time stimulating only the IPN terminals in the LDTg (Fig. 2e). ChR2 was expressed non-specifically in IPN neurons, and fiber optic implants were placed unilaterally above the LDTg (Fig. 2e, Supplementary Figs. 1, 4A). Photo-stimulation at 20 Hz of IPN terminals in the LDTg resulted in significant behavioral avoidance (Fig. 2g–i), consistent with the LDTg being an important target of the IPN in mediating aversion. Weaker, 1 Hz stimulation of IPN terminals in the LDTg did not induce behavioral avoidance, suggesting that strong activation of this connection is required to elicit aversion.

**Nicotine modulates IPN inputs to LDTg.** As high nicotine doses can activate the IPN[6], we tested whether nicotine could modulate IPN–LDTg synapses. ChR2 was expressed non-specifically in IPN neurons, and optically-evoked inhibitory post-synaptic currents (oIPSCs) were measured in VTA-projecting LDTg neurons (Fig. 3a, Supplementary Figs. 3A-D) as various nicotine concentrations (10 μM, 1 μM, and 100 nM) were applied to the slices.

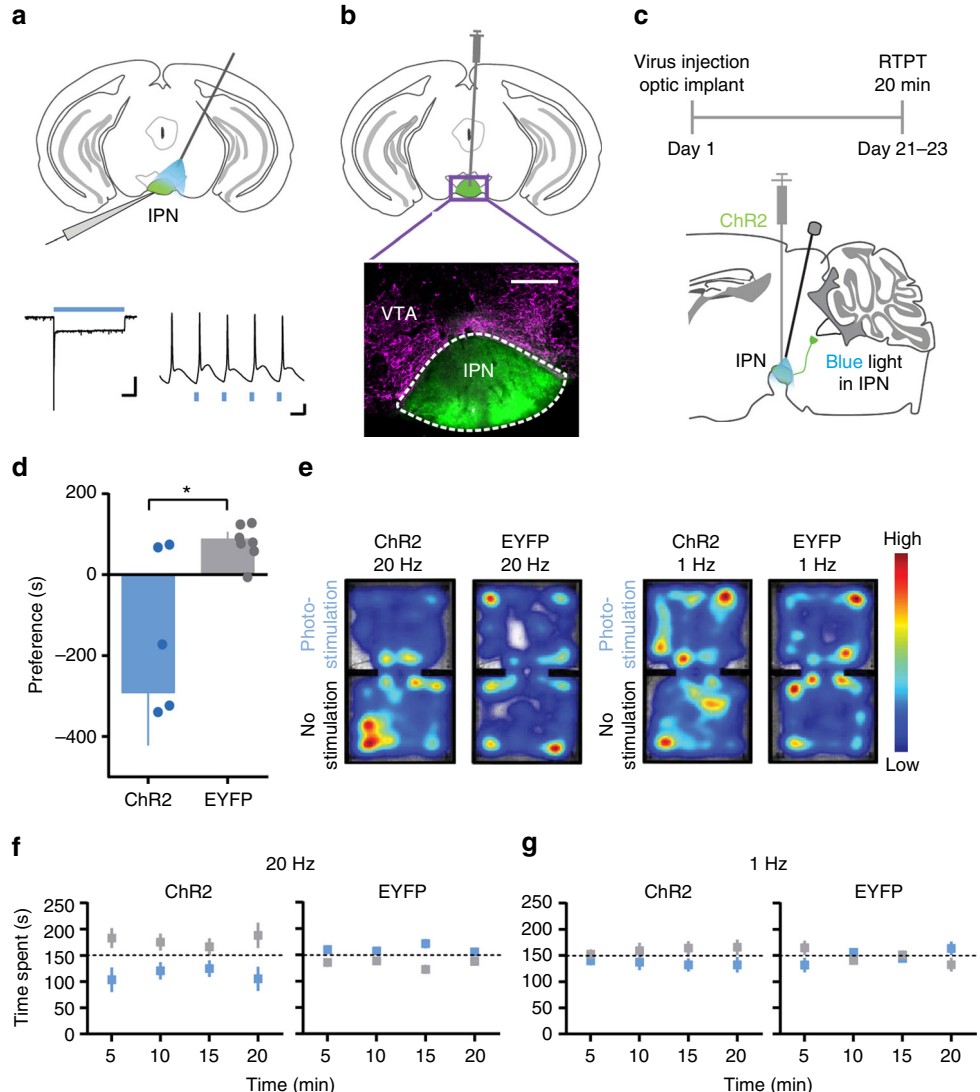

**Fig. 1** Optogenetic activation of IPN neurons elicits place avoidance. **a** Upper: Schematic of recording set-up; lower: light-evoked current (1 s, 100 pA scale) and action potentials in IPN neurons (50 ms, 10 mV scale). **b** Fluorescent image of IPN ChR2 expression (green) and proximal VTA (magenta TH staining) (250 μm scale). **c** Timeline and schematic of experimental set-up. **d** Preferences for photo-stimulation side vs. no stimulation side in RTPT with 20 Hz stimulation; unpaired $t$-test (two-tailed), $t_{12} = 2.922$, $P = 0.0128$, $n = 7$ mice per group. **e** Representative heat maps of animal positions during 20 Hz and 1 Hz stimulation. **f** Time spent (5 min bins) on photo-stimulation side (blue box) and no-stimulation side (grey box) at 20 Hz and **g** at 1 Hz. Data presented as mean ± SEM, * $P < 0.05$

Nicotine 10 μM was chosen to provide a strong activation of low-affinity nAChRs. A dose of 1 μM corresponds to serum levels of nicotine in mice following an aversive nicotine dose[43], and 100 nM nicotine corresponds to serum concentrations in smokers[44]. Both 10 μM and 1 μM nicotine significantly increased oIPSC amplitudes, while 100 nM nicotine did not (Fig. 3a, b, Supplementary Fig. 5D). Additionally, 10 μM nicotine resulted in a higher prevalence of nicotine-induced increases in eIPSC amplitude than 100 nM (Fig. 3c, Supplementary Figs. 5E-G). Thus, high aversive nicotine concentrations enhance the IPN–LDTg connection in our recordings, while a lower rewarding concentration does not.

These results are in line with our finding that strong activation of IPN inputs to LDTg promotes aversion, as high concentrations of nicotine that condition aversion also significantly activate this connection. Similarly, weaker, more tonic-like activation of IPN inputs to LDTg does not elicit aversion, and low concentrations of nicotine that do not correspond to aversion do not significantly impact this connection.

**Presynaptic β2 nAChRs mediate nicotine effects in LDTg.** Next, we investigated the mechanism by which nicotine modulates this synapse. Paired pulse ratios (PPR) of oIPSCs decreased after application of 1 μM nicotine (Fig. 3d, e), indicating that nicotine enhances release probability at this synapse. Miniature IPSCs (mIPSCs) recorded from back-labeled LDTg neurons show that 10 μM and 1 μM nicotine application significantly enhance mIPSC frequency, while 100 nM does not (Fig. 3f, g, Supplementary Figs. 6A-D).

The prevalence of nicotine-induced increases in mIPSC frequency was also higher for 10 μM than for 100 nM nicotine (Fig. 3h, Supplementary Figs. 6B-D). No nicotine concentration affected mIPSC amplitudes (Fig. 3i, Supplementary Fig. 7), and inclusion of 10 mM 1,2-bis(2-aminophenoxy)ethane-$N,N,N',N'$-tetraacetic acid (BAPTA) in the recording pipette did not block the nicotine-induced increase in mIPSC frequency (Fig. 3j, Supplementary Figs. 6A, E). All of these findings are consistent with activation of a low-affinity[13], pre-synaptic nAChR on IPN terminals in the LDTg.

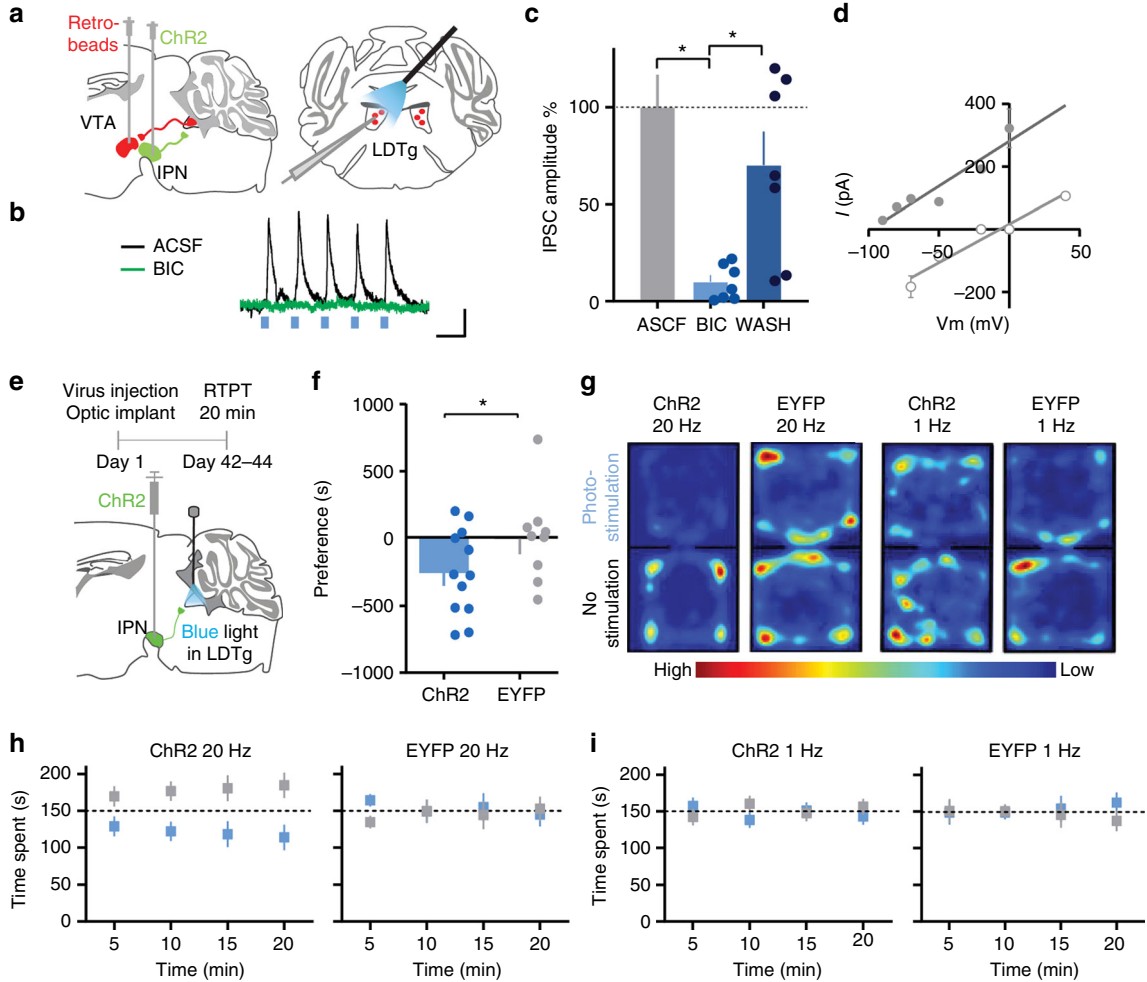

**Fig. 2** IPN inputs to LDTg are GABAergic and mediate aversion. **a** Left: schematic of viral and dye injections. Right: schematic of recording set-up. **b** Representative light-evoked synaptic currents before and during bicuculline (100 ms, 20 pA scale). **c** Normalized light-evoked synaptic current amplitudes before, during, and after bicuculline application (20 µM); one-way RM ANOVA, $F_{1.147, 6.882} = 7.863$, $P = 0.0246$; Tukey: $P = 0.0188$, $P = 0.0321$, $n = 7$ cells from 5 mice. **d** Current–voltage relationship, (grey circle) $E_{Cl} = -125$ mV, (white circle) : $E_{Cl} = -17$ mV. (linear regression, 95%CI: $-144.6$ to $-78.51$; 95%CI: $-23.78$ to 16.94; slopes: $P = 0.4098$; reversals: $-98.30$ mV and $-6.671$ mV, adjusted for junction potential: $-114.2$ mV and $-16$ mV; $n = 2$ cells per condition from 2 mice). **e** Behavioral experiment timeline and schematic. **f** RTPT preference scores for 20 Hz stimulation; unpaired $t$-test (one-tailed), $t_{19} = 1.774$, $P = 0.0460$, $n = 12$, $n = 9$ mice per group. **g** Representative heat maps of animal positions (20 Hz and 1 Hz stimulation). **h** Time spent on photo-stimulation side (blue box) and no-stimulation side (grey box) with 20 Hz stimulation. **i** Same as **h** but with 1 Hz stimulation. Data presented as mean ± SEM, * $P < 0.05$

We next investigated the subunit compositions of these nAChRs pharmacologically, testing nAChR antagonist effects on the 10 µM nicotine-induced increase in mIPSC frequency. The non-selective antagonist mecamylamine (MEC (*N*,2,3,3-tetramethylbicyclo[2.2.1]heptan-2-amine hydrochloride); 50 µM) blocked the increase, demonstrating that nAChRs mediate these effects (Fig. 2k, l, Supplementary Fig. 8B). The IPN has high α3β4 nAChR expression, which have a relatively low affinity for nicotine[26,45]. However, the selective α3β4 antagonist, SR-16584 (SR, 25 µM), had no effect on the nicotine-induced increase in mIPSC frequency (Fig. 3k,l), nor did the selective α7 nAChR antagonist, α-Bungarotoxin, (αBTX, 50 nM) (Fig. 3m, n, Supplementary Fig. 8C, D). Only bath application of dihydro-β-erythroidine hydrobromide (DhβE, 500 nM), the β2-containing nAChR antagonist, partially inhibited nicotinic enhancement of mIPSC frequency (Fig. 3k,l, Supplementary Fig. 8A), indicating that nAChRs on IPN terminals in the LDTg contain β2 subunits.

## Inhibition of IPN terminals in LDTg blocks nicotine CPA.
To test whether the IPN–LDTg connection is important for nicotine

CPA specifically, we inhibited IPN terminals in the LDTg using archaerhodopsin (ARCH; Fig. 4a, Supplementary Fig. 9D). ARCH was expressed non-specifically in IPN neurons, and expression was confirmed visually and electrophysiologically (Fig. 4a, Supplementary Fig. 1). We tested for blockade of nicotine-induced CPA following alternating conditioning sessions during which either an aversive nicotine dose (1.5 mg kg⁻¹) was paired with light stimulation on one side of the chamber or vehicle and no light delivery was paired with the opposite side (Fig. 4a). Preference scores (time spent on nicotine side−time spent on vehicle side) were normalized to the initial preference scores by subtracting the mean initial preference score of the group from each individual preference score. This conserves variability in pretest scores and yields positive numbers for preference and negative numbers for aversion in the posttest scores. Statistical analyses were done on the raw preference scores (Supplementary Fig. 9A), but normalized data are shown here for clarity. Using a two-way repeated-measures (RM) analysis of variance (ANOVA), we found no significant effect of time or group but did find a significant interaction between the two measures (Time:

$F_{1, 19} = 2.975$, $P = 0.1008$; Group: $F_{1, 19} = 0.01494$, $P = 0.9040$; Interaction: $F_{1, 19} = 9.109$, $P = 0.0071$). Bonferroni post-hoc test for multiple comparisons revealed that enhanced yellow fluorescent protein (EYFP) controls developed significant

aversion to the nicotine-paired side ($t_{19} = 3.842$, $P = 0.0022$), while ARCH mice on average did not ($t_{19} = 0.8218$, $P = 0.8428$) (Fig. 4b, c). Additionally, ARCH stimulation significantly attenuated the nicotine-induced conditioned aversion observed in the

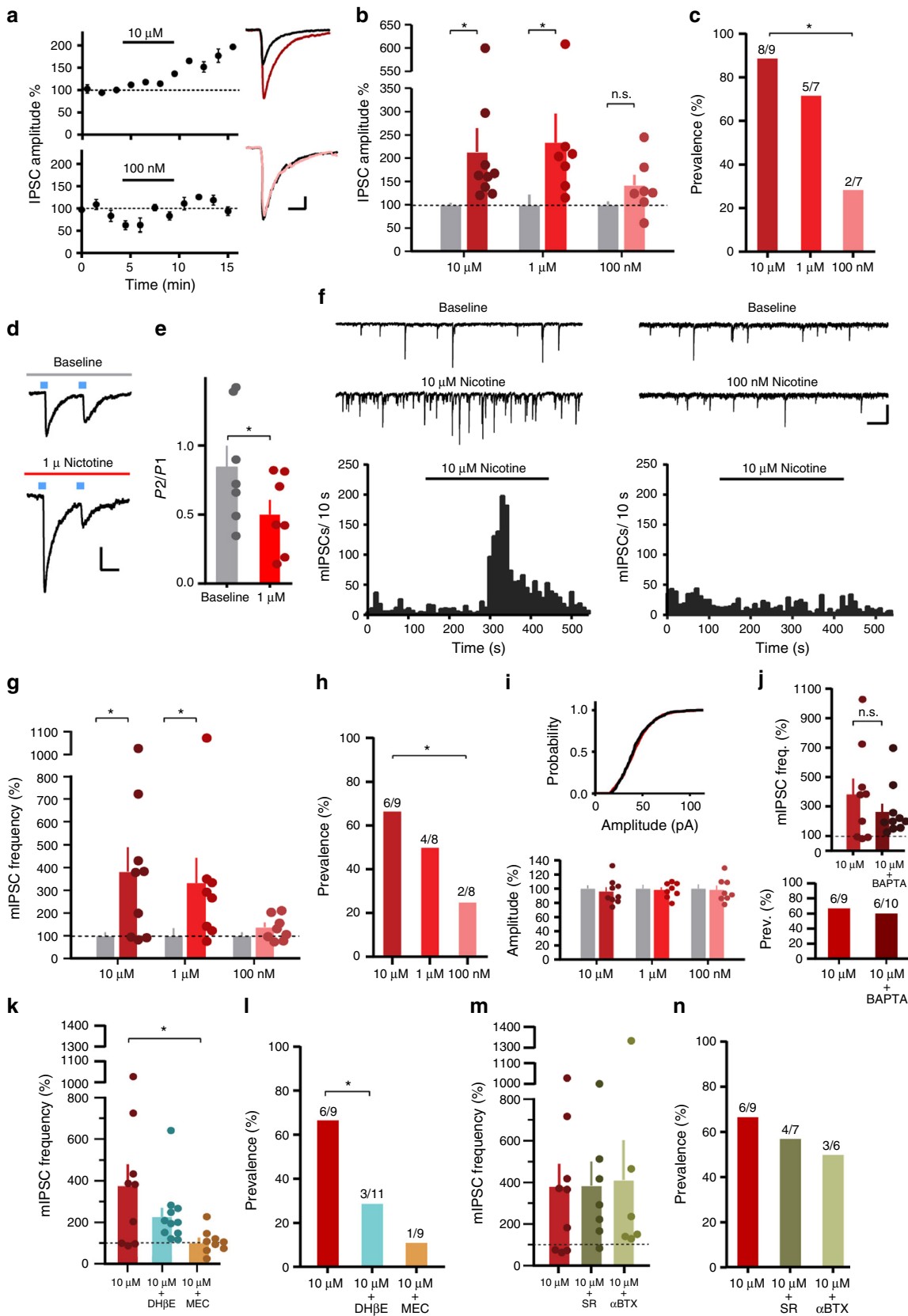

EYFP mice (Fig. 4d). These data support the idea that the IPN–LDTg connection mediates the aversive effects of high nicotine doses.

To control for the possibility that terminal inhibition alone was rewarding, we inhibited IPN terminals without nicotine and found no preference for the light-paired side (Supplementary Fig. 9B, C, E), indicating that terminal inhibition is not rewarding. In fact, ARCH mice tended to avoid the light-paired side, though this effect was not statistically significant (Supplementary Fig. 9B, C, E). This paradoxical trend led us to investigate the effects of ARCH activation on IPN–LDTg synaptic transmission. Spontaneous IPSCs (sIPSCs) were recorded in VTA-projecting LDTg neurons (Supplementary Fig. 9G-I). For the first 30 s, light delivery effectively inhibited sIPSCs, but at longer durations, sIPSC frequency increased, consistent with recent reports that ARCH terminal inhibition enhances spontaneous neurotransmitter release but inhibits evoked release[46,47]. Most IPN neurons do not fire tonically[48], so ARCH enhancement of spontaneous GABAergic release from IPN terminals onto LDTg cells elicits an aversive response. However, in the presence of high nicotine doses, IPN neuron firing rate increases[6], and we expect ARCH to effectively inhibit this action potential-dependent release. As the high nicotine dose also activates reward circuitry, the net effect is a shift in behavior away from aversion.

**Inhibition of IPN terminals in LDTg is anxiolytic**. Clearly, the IPN–LDTg connection did not evolve to mediate the aversive effects of nicotine. Nicotine can have anxiogenic effects, especially at high doses[2], so we tested whether this projection contributes to a general anxiety state using a modified light–dark box. IPN terminal inhibition in the LDTg with ARCH reduced latencies to enter the bright side of the apparatus (Fig. 4e). As the latencies were < 30 s, we are confident that ARCH effectively inhibited terminals during testing (Supplementary Fig. 9G-I). Thus, the IPN–LDTg connection contributes to the expression of anxiety, which may contribute to nicotine-induced aversion.

## Discussion

We have identified and clarified the nature of an important inhibitory IPN projection to the LDTg and demonstrated its behavioral relevance. Modulating this connection can shift the balance between reward and aversion, supporting the larger hypothesis that aversion can regulate nicotine-related behaviors[18,22,40]. Previous studies have focused on the MHb and its effects on the IPN as the mediator of nicotine aversion,

but the projection from IPN to LDTg is an important site where nicotine can alter synapses and behavior.

It should be noted that there are many subnuclei within the IPN, and each of these receives unique projections from the MHb and other regions, with distinct projections to different target regions in a cell-specific manner[21,33,34,49,50]. Our goal was to establish that GABAergic IPN projections to the LDTg play a role in the aversive effects of high doses of nicotine. To that end, we expressed ChR2 in the IPN in a non-subregion-specific manner to achieve high levels of expression in axon terminals within the LDTg, while avoiding expression in the nearby VTA (Supplementary Fig. 1). As a result, it is likely that ChR2 expression was lower in the rostral part of the IPN, which has been shown to contain specific cell populations that project to the LDTg[33,34,51]. While our studies did not focus on specific IPN cell types or subregions, it is possible that distinct subnuclei or cell-type-specific IPN projections to the LDTg mediate different behavioral end points, and this is an exciting direction for further understanding of this important pathway. In fact, one recent study has shown that a specific population of neurons that reside in the IPN mediates nicotine reward by inhibiting MHb terminals to reduce excitatory drive in the IPN and that these neurons project to the LDTg and the dorsal raphe[51]. However, it is unclear what consequences this cell-specific reduction in excitatory drive in the IPN has on signaling in projection regions. Given that the LDTg is a heterogeneous nucleus, it is also possible that different populations of IPN neurons target different cell types in the LDTg or target LDTg cells that project to different brain regions. We focused these initial studies on LDTg neurons that project to the VTA because activation of LDTg inputs to the VTA mediates reward[37], and LDTg connections to the VTA promote burst firing[36,52]. Given our findings, it is of great interest to further dissect these connections to determine the contribution of differential IPN innervation to specific LDTg cell types to the balance between nicotine reward and aversion.

While we have identified one nAChR subunit that contributes to the concentration-dependent effects of nicotine at the IPN–LDTg synapse, the precise subunit combinations that are relevant to aversion to high nicotine doses remain unknown. Our pharmacological approach was aimed at assessing receptor subunits implicated in nicotine reward and aversion as suggested by previous work[53,54]. The nAChRs identified here that modulate IPN–LDTg synapses have lower nicotine affinity than other β2-containing receptors expressed elsewhere. Identifying the other subunits of these receptors may provide novel therapeutic targets for addiction treatment, but this identification will be difficult

**Fig. 3** IPN inputs to LDTg are concentration-dependently modulated by nicotine via β2-containing nAChRs. **a** Left: example time course, normalized oIPSC amplitudes (30 ms, 50 pA scale). Right: representative traces (black: baseline, color: nicotine). **b** Normalized average oIPSC amplitudes, baseline (gray) vs. nicotine (colored); ratio paired $t$-tests (one-tailed), 10 μM: $t_8 = 3.492$, $P = 0.0041$, $n = 9$ cells from 6 mice; 1 μM: $t_6 = 3.367$, $P = 0.0075$, $n = 7$ cells from 3 mice; 100 nM: $t_6 = 1.708$, $P = 0.0692$, $n = 7$ cells from 3 mice. **c** Prevalence of increase in oIPSC amplitudes; chi-square (two-sided), $Z_{6.112, 1} = 2.472$, $P = 0.0134$, $n = 9$, $n = 7$ cells. **d** Representative PPR traces (50 ms, 25 pA scale). **e** Summary PPR graph; unpaired $t$-test (one-sided), $t_{12} = 1.804$, $P = 0.0482$, $n = 7$ cells from 3 mice. **f** Upper: Representative mIPSC traces (50 ms, 50 pA scale). Lower: Representative histograms of mIPSC frequency. **g** Normalized average mIPSC frequencies, baseline (gray) vs. nicotine (colored); ratio paired $t$-tests (one-tailed), 10 μM: $t_8 = 3.33$, $P = 0.0052$, $n = 9$ from 7 mice; 1 μM: $t_7 = 3.176$, $P = 0.0078$, $n = 8$ cells from 3 mice; 100 nM: $t_7 = 1.546$, $P = 0.0830$, $n = 8$ cells from 4 mice. **h** Prevalence of increase in mIPSC frequency; chi-square test (one-sided), $Z_{2.951, 1} = 1.718$, $P = 0.0429$, $n = 9$, $n = 8$ cells. **i** Upper: Representative cumulative amplitude histogram, baseline (black) vs. 10 μM nicotine (red). Lower: Normalized average mIPSC amplitudes, baseline (gray) vs. nicotine (colored). **j** Upper: 10 μM nicotine vs. 10 μM nicotine application + BAPTA in recording pipette; unpaired $t$-test (one-tailed), $t_{17} = 1.012$, $P = 0.1629$, $n = 9$ cells from 6 mice, $n = 10$ cells from 5 mice. Lower: Prevalence of mIPSC frequency increase; chi-square test (one-sided), $Z_{0.09048, 1} = 0.3008$, $P = 0.3818$, $n = 9$, $n = 10$ cells. **k** Normalized average mIPSC frequencies, 10 μM nicotine vs. 10 μM nicotine + DhβE or MEC; one-way ANOVA, $y = \log(y)$ transform, $F_{2, 26} = 6.175$, $P = 0.0064$ (main effect), Holm–Sidak P = 0.3830, P = 0.0048, for DhβE or MEC, respectively. **l** Prevalence of mIPSC frequency increase; chi-square test (one-sided), $Z_{3.104, 1} = 1.762$, $P = 0.0391$, $n = 9$ cells from 2 mice, $n = 11$ cells from 3 mice. **m** Normalized average mIPSC frequencies, 10 μM nicotine vs. 10 μM + SR or 10 μM + αBTX; $n = 9$, $n = 7$ cells from 2 mice, $n = 6$ cells from 2 mice. **n** Prevalence of mIPSC frequency increase. Data presented as mean ± SEM, * $P < 0.05$

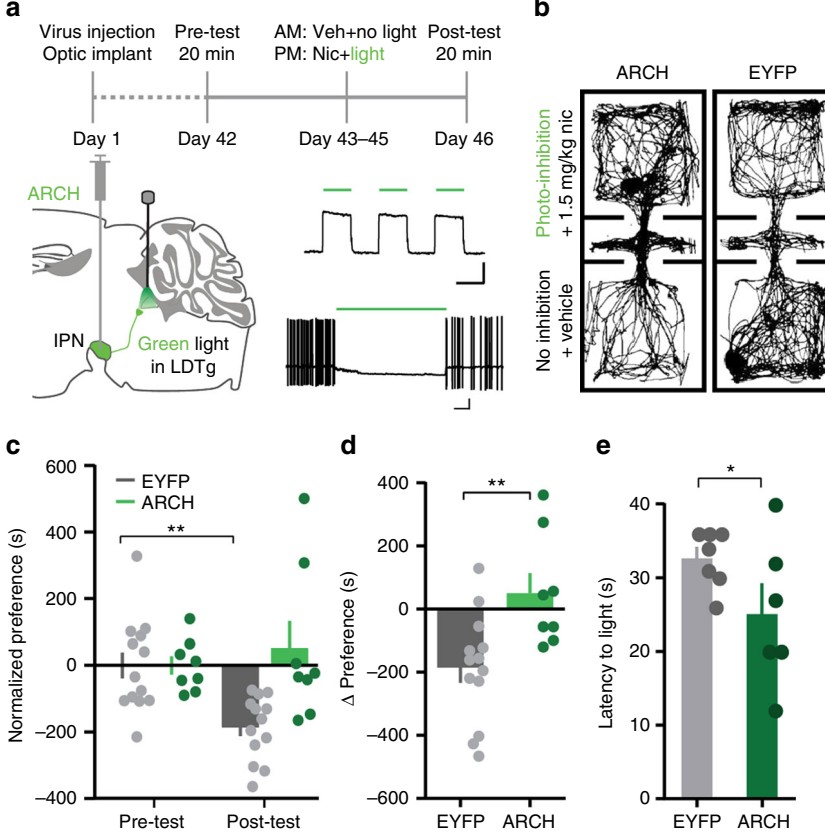

**Fig. 4** Inhibition of IPN terminals in LDTg blocks nicotine conditioned place aversion. **a** Upper: experimental timeline. Lower left: schematic of viral expression and fiber implants. Right: functional expression of ARCH, (500 ms, 100 pA; 2 sec, 5 mV scale). **b** Representative plots of animal position during post-test. **c** Normalized average preference scores; two-way RM ANOVA with Bonferroni post-hoc: $P = 0.0022$, EYFP $n = 13$ mice; ARCH $n = 8$ mice. **d** Change in preference for nicotine-paired side after conditioning: unpaired $t$-test (two-tailed), $t_{19} = 3.018$, $P = 0.0071$. **e** Latency to enter bright side; unpaired $t$-test (one-tailed), $t_{11} = 1.858$, $P = 0.0450$, EYFP $n = 7$, ARCH $n = 6$ mice. Data presented as mean ± SEM, ** $P < 0.01$, * $P < 0.05$

using pharmacology alone given the lack of subunit-selective agonists and antagonists, especially for α5 and α2 subunits, and genetic tools will be required to fully elucidate this detailed information[53,54].

Nicotine withdrawal is also mediated by the MHb–IPN pathway[11], and future work should investigate the contribution of the IPN–LDTg connection to the aversive aspects of withdrawal. Our findings suggest that this pathway might be especially relevant for withdrawal-induced anxiety, which has already been investigated at the level of the IPN[29,49,50,55]. It is interesting to consider the relationship between anxiety and aversion more generally. Recent work has focused on understanding the role of the IPN in modulating reward-related behaviors but not via aversion per se. One such study shows that inhibiting GABA neurons in the IPN can enhance the rewarding properties of a familiar encounter, while activating these neurons reduces the value of a novel encounter[49]. The novel encounter never becomes aversive, but it seems that the IPN activity shifts the motivational valence of the novel encounter toward the less rewarding valence of a familiar encounter. Another study has shown that GLP-1 signaling in the IPN can control nicotine self-administration by altering excitatory drive from the MHb[28]. As in the familiarity study, GLP-1 signaling in the IPN is not aversive per se, but it does limit nicotine intake. Because GLP-1 is also involved in feeding and satiety, the authors suggest a satiety-like mechanism for limiting nicotine intake via GLP-1 signaling in the IPN. The IPN has also been implicated in fear learning[56,57]. More work is clearly required to fully understand the role that the IPN plays in these various but related behavioral states.

## Methods

**Animals**. All experiments were done with the approval of the University of Chicago's Institutional Animal Care and Use Committee. Adult male (>8 week old) C57/Bl6 mice (Jackson Laboratories or bred in-house) were group housed in a colony room on a standard light–dark cycle (6 A.M.–6 P.M.). Upon arrival, mice were undisturbed for at least 72 h to allow acclimatization to the colony room. Water and standard chow were available ad libitum, and cages were changed twice per week. Experiments were conducted during the day, during the animals' light period.

**Drugs and reagents**. All chemicals were obtained from Sigma Aldrich unless otherwise indicated: nicotine (nicotine hydrogen tartrate salt), DNQX (6,7-dinitroquinoxaline-2,3-dione, ABCAM), bicuculline (Tocris), TTX (tetrodotoxin citrate, ABCAM), BAPTA (1,2-bis(2-aminophenoxy)ethane-N,N,N',N'-tetraacetic acid, Tocris), SR-16584 (1,3-dihydro-1-(3-exo)-9-methyl-9-azabicyclo[3.3.1]non-3-yl]-2H-indol-2-one, Tocris), MEC (mecamylamine, Tocris), αBTX (α-Bungarotoxin, Tocris), and DhβE (di-hydro-β-erithroidine hydrobromide, Tocris).

**Surgical procedures and viruses used**. All mice were at least 8 weeks of age before undergoing surgery. Anesthesia was induced and maintained with isofluorane at 4 and 1%, respectively. Mice were placed in a stereotaxic frame, and craniotomies were performed for brain injections. AAV2-hSyn-hChR2(H134R)-EYFP, AAV2-hSyn-EYFP, and AAV-hSyn-eArch3.0-EYFP were obtained from the University of North Carolina vector core and were injected directly into the IPN at a volume of 250 nl (AP: −3.5 mm, ML: −1.0 mm, DV: −4.92 mm from Bregma at a 10° angle). These coordinates target the ventral IPN to avoid infection of the nearby VTA, but in many cases, most of the IPN expressed ChR2 (Fig. 1b, Supplementary Fig. 1).

For behavioral experiments, fiber optics were permanently implanted either above the IPN (AP: −3.5 mm, ML: −1.0 mm, DV: −4.36 mm from Bregma at a 10° angle) (Fig. 1c, Supplementary Fig. 2A) or unilaterally above the LDTg (AP: −5.25 mm, ML: + or −0.4 mm, DV: −3.4 mm from Bregma) (Fig. 2e, Supplementary Figs. 4A, 9D). These coordinates target the caudal LDTg to avoid IPN terminals in the nearby dorsal raphe nucleus. Dental acrylic (Lang Dental, Wheeling, IL) was used to secure the fiber optic implants[58].

For electrophysiology experiments, retrograde labeling was accomplished by injecting fluorescent microspheres (FluoSpheres, Life Technologies) bilaterally into the VTA (AP: −3.0 mm, ML: +/−0.5 mm, DV: −3.0 mm from bregma) (Fig. 2a, Supplementary Fig. 3A, B). For behavioral and electrophysiological experiments requiring direct light stimulation of the IPN, animals were allowed to recover for at least 3 weeks before experiments were conducted to allow adequate somatic protein expression. For behavioral and electrophysiological experiments requiring light stimulation of IPN terminals, animals were allowed to recover for at least 6 weeks before experiments were conducted to ensure adequate protein expression in the terminals.

**Slice preparation.** Mice were rapidly decapitated following anesthesia with iso-flurane (Baxter, Deerfield, IL). Brain slices were obtained using a neuroprotective recovery method adapted from ref. [59]. Briefly, brains were dissected in a solution of ice-cold protective artificial cerebrospinal fluid (aCSF) including: in mM, 92 N-methyl-D-glucamine, 2.5 KCl, 1.25 NaH$_2$PO$_4$, 30 NaHCO$_3$, 25 glucose, 2 thiourea, 5 Na-ascorbate, 3 Na-pyruvate, 0.5 CaCl$_2$·4H$_2$O, and 10 MgSO$_4$·7H$_2$O, pH adjusted to 7.3–7.4 with HCl and then bubbled continuously with 95% O$_2$–5% CO$_2$. 250-μm-thick sagittal or coronal slices containing IPN or LDTg were cut with a vibratome (VT100S, Leica) and incubated in a holding chamber at 32–34 °C for ≤15–20 min in the same protective fluid. Slices were then transferred to a second holding chamber containing room temperature aCSF including: in mM, 119 NaCl, 2.5 KCl, 1.25 NaH$_2$PO$_4$, 26 NaHCO$_3$, 12.5 glucose, 2 CaCl$_2$·4H$_2$O, 2 MgSO$_4$·7H$_2$O, 2 thiourea, 5 Na-ascorbate, 3 Na-pyruvate, 20 HEPES, bubbled continuously with 95% O$_2$–5% CO$_2$ and perfused at a rate of 20 ml min$^{-1}$ for at least 30 min before recording.

**Slice electrophysiology.** Recording chambers were superfused (~2 ml min$^{-1}$) with room temperature aCSF (in mM, 125 NaCl, 25 NaHCO$_3$, 20 glucose, 2.5 KCl, 2.5 CaCl$_2$, 1 MgCl$_2$, 1 NaH$_2$PO$_4$, at pH 7.4, saturated with 95% O$_2$ and 5% CO$_2$). Neurons were visualized under infrared illumination using a fixed-stage upright microscope (Axioskop, Zeiss). Data were acquired with a Multiclamp 700A/Axo-patch 200B amplifier and pCLAMP 9 software (Molecular Devices). Whole-cell patch-clamp recordings were achieved with microelectrodes (3–6 MΩ) pulled on a Flaming/Brown micropipette puller (model P-97, Sutter Instrument, Novato, CA).

All electrophysiology experiments were performed on back-labeled neurons in the LDTg or on neurons in the IPN that expressed either ChR2 or ARCH (verified by EYFP fluorescence). Retrogradely labeled or virally infected neurons were visualized under fluorescence and bright field illumination (Supplementary Fig. 3C).

Recording electrodes were filled with either a low Cl$^-$ ($E_{Cl}$ −125 mV) potassium gluconate internal solution (in mM, 154 K-gluconate, 1 KCl, 1 EGTA, 10 HEPES, 10 glucose, 5 ATP, 0.1 GTP, pH 7.4 with KOH) or an intermediate Cl$^-$ ($E_{Cl}$ −17 mV) potassium gluconate internal solution (in mM, 70 K-Gluconate, 70 KCl, 1 EGTA, 10 HEPES, 10 Glucose, 5 ATP, 0.25 GTP, 15 sucrose, pH 7.4 with KOH) (Fig. 2d).

To activate light-sensitive proteins, light was delivered through the objective at maximal power (>40 mW; 473 nm or 532 nm). Except for the experiments shown in Fig. 2a, d, DNQX (100 μM) was included in the aCSF to block AMPA-mediated currents during all experiments during which GABAergic currents were measured. When blocking GABAergic currents, bicuculline (20 μM) was included in the aCSF. To record mIPSCs, TTX (1 μM) was included in the aCSF. To determine the nAChR subunit compositions at IPN terminals, nicotinic antagonists were included in the aCSF at the following concentrations: 50 μM MEC, 25 μM SR-16584, 50 nM α-Bungarotoxin, 500 nM DhβE. Nicotine was bath-applied at concentrations of 10 μM, 1 μM, or 100 nM. Data were only included from recordings with series resistance <30 MΩ, and where input resistance or series resistance varied by <25% throughout the recording. Current-clamp recordings were all done at $I = 0$.

**Histology.** *For electrophysiology experiments*: After recordings, slices were trans-ferred to 4% paraformaldehyde for at least 24 h. Correct localization of dye was confirmed by conducting immunohistochemistry. A rabbit antibody to tyrosine hydroxylase (1:500, Thermo OPA1-04050)[60] was used to identify dopamine neu-rons and indicate the boundaries of the VTA (Fig. 1b, Supplementary Fig. 3B). If the majority of the dye was found to be outside of the VTA, the data recorded from that animal was excluded (Supplementary Fig. 1).

Correct localization of viral infection was confirmed by immunohistochemistry as well. A chicken antibody to green fluorescent protein (which also detects EYFP) (1:5000, Abcam Ab13970)[61] was used to enhance the fluorescence for visualization of EYFP expression. If the expression of EYFP was not confined to the IPN without impinging on the proximal VTA, data recorded from that animal was excluded (Supplementary Fig. 1).

In a subset of slices, immunohistochemistry to visualize acetylcholine-expressing neurons was performed in an effort to confirm that recordings from back-labeled neurons were indeed conducted in the LDTg. A goat antibody to choline acetyltransferase (1:1000, Millipore Ab144)[62] was used to this end (Supplementary Fig. 3C, D).

*For behavioral experiments*: Animals were anesthetized with isofluorane and transcardially perfused with 4% paraformaldehyde. Brains were kept in paraformaldehyde for >24 h and then transferred to 30% sucrose in phosphate-buffered saline (PBS) for >24 h. Brains were frozen in embedding medium (OCT Compund, Tissue-Tek, Sakura Finetechnical) and 50 μm slices were taken using a cryostat (Leica CS3050 S). Fiber optic and viral expression were confirmed using anatomical markers[63]. Animals with incorrect placement of either fiber optics or viral injections were excluded from analysis (Supplementary Fig. 1, 2A, 4A, 9D).

**Behavioral testing.** Animals were habituated to the experimenter, the behavior room, and handling for 3–5 days prior to the start of experiments. Mice were connected to an optical fiber connected to a laser during this habituation period. For ChR2 experiments, blue light (473 nm) was delivered through the fiber optic implants at a frequency of either 20 Hz or 1 Hz. For ARCH experiments, green light (532 nm) was delivered continuously. All light was delivered at 7–12 mW power.

*Real-time preference test (RTPT)*: Mice were placed into a custom-made black acrylic, two-chambered box (52.5 × 25.5 × 25.5 cm³) and allowed to explore each of the two chambers for 20 min[64–66]. Using Noldus Ethovision hardware controller connected to a master 9 function generator, light stimulation was delivered through fiber optic implants during the duration of time the mouse spent in the light-paired side of the chamber. Light stimulation was stopped when the animal returned to the side paired with no light. Light was delivered in 10 ms pulses at either 20 Hz (as in Fig. 1a, lower right) or 1 Hz. The experimental animals were counterbalanced for both group and light-delivery side. Preference or aversion in each experiment was determined by subtracting the amount of time spent in the no stimulation side from the time spent in the photo-stimulation side during this real-time testing (Figs. 1 and 2). The same animals received both 20 Hz and 1 Hz light delivery on separate testing days. Frequencies were counter-balanced such that animals randomly received one stimulation on the first test day and the other stimulation on the second day. All behavioral data were analyzed using Noldus Ethovision (v11).

*Conditioned place aversion (CPA)*: Mice were trained in an unbiased, balanced three-compartment conditioning apparatus as described in refs. [67,68]. The compartmentalized box is divided into two equal-sized outer sections joined by a small center compartment accessed through a single doorway on each side. The compartments differed in wall striping (vertical vs. horizontal, alternating black and white lines). All exposures to the apparatus were recorded with a video camera and analyzed using Ethovision 11 (Noldus). Any time that the animals were exposed to the conditioning apparatus, they were connected to the fiber optic cables.

We randomly divided animals into four groups: ARCH+nicotine, EYFP +nicotine (Fig. 4a, d), ARCH+vehicle, and EYFP+vehicle (Supplementary Fig. 9B, C, E). On the pre-conditioning day (day 1), mice were allowed free access to all three chambers for 20 min. Mice were assigned vehicle and nicotine compartments based on their initial preference during the pre-conditioning day. We used a biased design, wherein nicotine was paired with the initially preferred side, and vehicle was paired with the initially less preferred side. This design has been reported to be the best method for assessing expression of nicotine conditioned preference or aversion in rodents[43]. For mice that received only vehicle injections, light was paired with the initially preferred side, to mimic the conditions of the nicotine groups. Mice received a vehicle (filtered PBS) injection in the morning (10 ml kg$^{-1}$, intraperitoneal (i.p.)) paired with no light (no inhibition), and they received a nicotine injection (1.5 mg kg$^{-1}$, i.p. as base) in the afternoon paired with light delivery (photo-inhibition) for the duration of time in the conditioning chamber (20 min, continuous light). The nicotine+light treatment was at least 4 h after the morning training, and this conditioning paradigm was repeated on 3 consecutive days. Nicotine was always administered during the afternoon session to increase the time between the last nicotine dose and the next conditioning session. This limits the possible confound of residual nicotine in the system influencing the conditioning. Pre- and post-conditioning testing were done at midday to avoid any confounds related to testing at the same time as only one of the conditions. On the fifth day, the mice were placed in the apparatus and allowed free access to the three compartments for 20 min. Pre- and post-conditioning sessions were videotaped for analysis of CPA. For the vehicle control experiment (Supplementary Fig. 9B, C, E), animals received vehicle injections during both photo-inhibition and no inhibition conditioning sessions.

*Light–dark box experiments:* Mice were habituated to the experimenter and to fiber optic connection. The apparatus consisted of two compartments with no distinctive contextual cues. One side was covered (dark side), resulting in light intensity of ~10 lx, and the other side was exposed to light (bright side) so that light intensity was ~400 lx inside the compartment. Animals were placed in the dark

side, and photo-inhibition was delivered continuously. Latency to enter the bright side was measured using Ethovision (Fig. 4E).

In all behavioral experiments, animals that exhibited any signs of illness were excluded from analysis. Additionally, animals were sometimes excluded owing to loss of fiber optic implants.

**Data analysis**. In all experiments, outliers, as determined by the ROUT method[69] ($Q = 0.1\%$), were removed from further analysis.

mIPSCs were analyzed with Mini-Analysis (Synaptosoft, Decatur, GA). Amplitude and area thresholds were used to acquire events and each event was visually inspected to protect against software errors. The number of events per 10 s bin was assessed, and unpaired $t$-tests (one-tailed) were used to identify significant frequency differences between baseline (100 s prior to nicotine application) and nicotine application periods (at least a 30 s duration) (Supplementary Fig. 6B-E, 8). The number of cells in which the frequency was significantly increased during nicotine application compared to baseline was used to determine the prevalence of a nicotine-induced increase (Fig. 3h, j, l, n). Chi-square tests were used to assess differences in prevalence of nicotine effects between groups. Baseline (ten 10 s bins) frequencies and nicotine (at least three 10 s bins) frequencies were averaged for each cell. These values were used in comparisons within groups after being normalized to baseline. Ratio paired $t$-tests (one-tailed) were then used to determine differences between nicotine and baseline frequencies within groups (Fig. 3j, Supplementary Fig. 6A). One-way ANOVA, or unpaired $t$-test if only comparing two groups, was used to determine differences in the magnitude of nicotine's effect between groups, and if the variance differed between groups, a $y = \log(y)$ transform was used (Fig. 3k, j), followed by a Holm–Sidak post-hoc test as appropriate. Changes in amplitude were tested using Kolmogorov–Smirnov tests on cumulative amplitude probability histograms (Supplementary Fig. 7). Baseline and nicotine periods were the same for frequency and amplitude analyses.

sIPSCs were analyzed in the same way as mIPSCs. Baseline was defined as the 100 s prior to light delivery, and light delivery effects were divided into two groups: the first 30 s of light delivery and the last 60 s of 5 min of light delivery. Within cells, unpaired $t$-tests (one-tailed) were done for baseline vs. first 30 s and baseline vs. last 60 s of light to determine the prevalence of effects of light delivery (Supplementary Fig. 9G). A paired $t$-test (one-tailed) was then used to determine whether, on average, the longer duration of light exposure enhanced sIPSC frequency (Supplementary Fig. 9I).

Light-evoked IPSCs were differentiated from failures by the same amplitude criteria used in the analysis of spontaneous transmission: deflections from baseline were required to have an onset consistent with light-delivery onset, to be >5× root mean square noise, and to have appropriate rise and decay characteristics in order to be considered evoked synaptic currents. The amplitude, rise time, and decay time of the oIPSCs were determined in real time by the pCLAMP 9 software (Axon Instruments). Care was taken in all electrophysiological studies to ensure that the holding current and series resistance were stable through the entire experiment. Every oIPSC was visually inspected to ensure that the software determined the parameters correctly. oIPSCs were evoked every 30 s.

RM one-way ANOVA was used to assess whether bicuculline effectively blocked oIPSCs and could be washed out (Fig. 2c). Linear regression was used to fit lines to the data points obtained from the current–voltage relationship experiment and to determine whether their intercepts or slopes were different from each other (Fig. 1d).

Unpaired $t$-tests (one-tailed) were used to identify significant light-evoked amplitude differences between baseline periods (five oIPSCs prior to nicotine application) and nicotine application periods (at least three oIPSCs) (Supplementary Fig. 3E-G). The number of cells in which the amplitude was significantly increased during nicotine application compared to baseline was used to determine the prevalence of a nicotine-induced increase (Fig. 2c). Chi-square tests were used to assess differences in the prevalence of nicotine effects between groups. Average amplitude values were used in comparisons between groups after being normalized to baseline. Ratio paired $t$-tests (one-tailed) were then used to determine differences between nicotine and baseline oIPSC amplitudes within groups (Fig. 2b, Supplementary Fig. 3D). A new slice was used for each experiment in which nicotine was applied so that each slice was only exposed to nicotine once.

During behavioral experiments, animals were recorded on video, and movements were analyzed using Ethovision software (Noldus). For RTPT, time spent in each side of the apparatus was recorded, as was gross locomotion (Supplementary Fig. 2B, 4B 2B, 4B). Unpaired $t$-tests were used to assess differences in preference scores between groups. During the first experiment when we stimulated the IPN directly, two-tailed test was used (Fig. 1d). When we followed up with IPN terminal stimulation in the LDTg, one-tailed test was used, as we predicted that, if there was a behavioral effect, it would be in the same direction as for the previous experiment (Fig. 2f). To test for any effects of these manipulations on locomotion, one-way ANOVA with Holm–Sidak post-hoc test was used (Supplementary Fig. 2B, 4B).

For CPA, time spent in each side of the apparatus was recorded, as was gross locomotion (Supplementary Fig. 9F). Two-way RM ANOVA was used to analyze this data. Time (pre- vs. post-conditioning) and Group (ARCH vs. EYFP) were the two factors. Bonferroni post-hoc analysis was used as appropriate (Fig. 4c, Supplementary Figs. 9A-C). To determine the difference in the change in

preference between these two groups, unpaired $t$-test (two-tailed) was used (Fig. 4d, Supplementary Fig. 9E). Nicotine and vehicle cohorts were analyzed separately. To test for any effects of these manipulations on locomotion, one-way ANOVA with Holm–Sidak post-hoc test was used (Supplementary Fig. 9F).

No statistical methods were used to predetermine sample sizes, but our sample sizes are similar to those reported in previous publications.

**Randomization and blinding**. Mice were not selected for any experimental condition based on previous observations or tests. Cages were selected arbitrarily to receive control or experimental viral injections. Individual mice were arbitrarily given numbers and tail marks reflecting those numbers. Behavior boxes were numbered left to right, starting with box one, and the animal number determined which box the mouse experienced throughout testing. Animal numbers also determined the order of testing. For example, ChR2 mice 1 and 2 would be tested at the same time as EYFP mice 1 and 2, and this would occur just prior to the session that would include ChR2 3 and 4 and EYFP 3 and 4. Care was taken that each box had equal numbers of control and experimental animals assigned to it over the course of an experiment (i.e., box 1 would not be assigned to only ChR2 mice). Light-stimulation frequencies were arbitrarily assigned along the same lines, with care taken that the same number of mice received each stimulation between groups and between days. Nicotine vs. saline conditions were assigned arbitrarily as well.

Behavioral tests and electrophysiological data acquisition were performed by investigators with knowledge of the experimental groups. All behavioral experiments were controlled by computer systems, and data were collected and analyzed in an automated and unbiased way. Histological verifications always took place prior to analysis of behavioral data. Experimenters were not blinded to the groups during this verification step but were blinded to the actual observed behavior of individuals and groups.

**Data availability**. All relevant data are available from the authors upon reasonable request.

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

## Acknowledgements

We thank Nichole M. Neugebauer for her help in editing the manuscript. This work was supported by NIH grants DA036978 (to D.S.M.), DA033396 (to M.R.B.), and DA038725 (to R.A.-H.)

## Author contributions

S.L.W., D.F.G., F.B., and K.L. collected data. S.L.W., R.A.-H., J.G.M., M.R.B., and D.S.M. designed experiments. S.L.W. and D.S.M. analyzed the data, and S.L.W., R.A.-H., J.G.M., M.R.B., and D.S.M. wrote the manuscript.

## Additional information

**Competing interests:** M.R.B. is a co-founder, and scientific advisor of Neurolux, Inc, a company invested in developing neuroscience tools and applications. None of the research reported here was supported by Neurolux, Inc. The remaining authors declare no competing interests.

