## [Peer Review File · Nature Communications]

Reviewers' comments:

Reviewer #1 (Remarks to the Author):

There is increasing interest in the interpeduncular nucleus (IPN) as it is emerging as a key brain area involved in addiction particularly in nicotine dependence. However, IPN efferents have not been mapped in much functional or behavioral detail. Wolfman et al. have, for the first time, shown that IPN GABAergic neurons functionally innervate the lateral dorsal tegmental area (LDTg). Using optogenetics, they find that stimulation of the IPN, or the specific IPN terminals that innervate the LDTg, induce aversive behaviors by increasing GABAergic inhibition of LDTg neurons that project to the ventral tegmental area. As activation of the IPN underlies the aversive properties of high nicotine doses, they go on to identify the mechanism by which high nicotine concentrations increase inhibition of LDTg neurons and identify the nicotinic acetylcholine receptor subtype involved. Finally, they test their hypothesis by photoinhibiting IPN->LDTg terminal during high dose nicotine exposure to block aversion. Ultimately, they conclude that nicotine aversion is mediated by GABAergic IPN inputs to the LDTg. Overall, the experiments are well done, the data are robust, and the conclusions are justified. My specific comments are:

1. Supplementary figure 1 seems like a critical part of the manuscript and should be moved from the supplement to the main text (i.e. it should be figure 1) since the effect of direct, non-specific activation of the IPN by channelrhodopsin on mouse behavior has not been shown before.
2. It would be helpful if images showing the projection from the IPN to the LDTg, taken from mice expressing Chr2 in the IPN, were included in the manuscript (or at least YFP expression in the LDTg after virus injection in the IPN).
3. Figure 3C is difficult to understand because the groups are not labeled and a key for the colors is not included. Also, data for this experiment should be analyzed by two-way ANOVA with test day and virus expression as main factors.
4. It appears as if the manuscript was formatted for a short communication. In addition to point 1 above, the discussion should be expanded to include how the new data fits with what is known regarding the IPN based on recent literature (i.e., Ables et al., 2017 PNAS, Koppensteiner et al., 2017 Cell Reports; Molas et al., Nat Neurosci., 2017; etc.).

Minor:

1. Supplementary figure 8 appears out of place and should be moved to the beginning of the supplementary data.

Reviewer #2 (Remarks to the Author):

This study, by Wolfman et al. investigates the role of the interpeduncular nucleus laterodorsal tegmentum (IPN-LDTg) neuronal projection in aversion to nicotine in mice.

The strengths of the paper are: 1) a behavioral task (real time preference test/conditioned place preference) that measures place aversion and allows to combine optogenetic manipulations with nicotine pairing 2) electrophysiological recordings in IPN and IPN-LDTg at different nicotine doses 3) Combined use of retrobeads to visualize VTA projecting LDTg neurons and fluorescent tagged Channelrhodopsin (ChrR2) or archaerhodopsin (ARCH3) to optogenetically manipulate IPN projections to the LDTg.

As a result, there are a number of remarkable findings in this study. First, the authors show that

optogenetic activation of IPN neurons, achieved by blue light ChR2 activation of either the cell bodies in the IPN or of the terminals projecting to the LDTg, increases place aversion. The authors next tested whether this optogenetic activation could be modulated by nicotine, and found that only high aversive doses of nicotine enhance the IPN-LDTg transmission by increasing the release probability and mIPSC frequency (presynaptic mechanism). Using different nAChR blockers they conclude that this effect is mediated by $\beta 2$ containing nAChR but could not identify the alpha subtype/s. Finally, the authors use ARCH3 to inhibit the IPN-LDTg connection. They show that photoinhibition alone produces aversion, that an aversive dose of nicotine also produces place aversion but that photoinhibition paired with a high dose of nicotine does not produce aversion. These results are a bit counterintuitive since ARCH activation paradoxically increases spontaneous release but blocks evoked release.

Overall, the results are compelling, data are presented in a manner that is generally easy to understand, interpretations are logical, and the manuscript is well written. This study is highly relevant to the understanding of the role of the IPN-LDTg to nicotine aversion, and has the potential to be a great paper.

I have the following criticisms, which I hope will strengthen the manuscript.

Main comments

1. Cell-population specificity is missing. ChR2 and ARCH3 were not delivered in a cell specific manner using Cre-mice or similar. Viral injections were done in the ventral/caudal area of the IPN and avoided the rostral part of the IPN (IPR) to prevent infection of the VTA. However the IPR contains a specific cell population that also projects to the LDTg that expresses $\alpha 5$ (Hsu et al. J.Neurosci 2013), $\alpha 4$ and somatostatin (Zhao-Shea et al, Curr Biology 2013) and NOS (Ables et al. PNAS 2017) that may have an opposite role in nicotine aversion (Ables 2017) Also the intermediate IPI seems to contain a specific cell population that when activated increases anxiety (Zhao-Shea Nat comm 2014). This lack of cell specificity should be discussed and maybe it should be mentioned in the abstract that mostly the ventral part of the IPN was targeted as it is possible that different populations in the IPN might not cause aversion to nicotine.
2. The studies that identified $\beta 2$ nAChR presynaptic receptors in the IPN-LDTg connection did not identify the alpha subunit. Could the authors discuss or test more blockers to test the possibility that $\alpha 5$, $\alpha 2$ or other alpha subunits could be involved in the response to nicotine.
3. The authors pair optogenetic manipulations with nicotine at different doses (2 aversive doses and a lower dose found in serum levels of smokers). The lower concentration found in smokers does not affect ChR2 light-evoked IPSC amplitude or frequency. And ARCH was also paired with only a high aversive nicotine dose. Also the lower photostimulation of 1 Hz does not affect place aversion. How these high nicotine doses and photostimulation frequencies relate to physiological levels or levels found in smokers. Could the authors describe this in more detail.

Reviewer #3 (Remarks to the Author):

In this manuscript, the authors have examined a projection from the interpeduncular nucleus (IPN) to the lateraldorsal tegmentum (LDTg). With optogenetic approaches and behavioral assessments, these studies suggest that activation of the IPN elicits an aversive response in a real-time place preference behavioral assessment, an effect likely mediated through downstream modulation of LDTg neurons projecting to the VTA. The doses of nicotine examined have been appropriately rationalized as levels relative to that found in human smokers. Delineation of the function of this pathway is novel and of high interest to the community and wider field. However, significant concerns are noted which limit the strength of the conclusions. Specific comments are as follows:

Major

1. Overall, the behavioral data are not very convincing as significant overlap is found between groups, with outliers likely contributing to the statistical differences found with one-tailed t-tests. For instance, with regard to Figure 3, the authors conclude that 'EYFP controls developed significant aversion to the nicotine-paired side, while ARCH mice did not'. However, more than half of the ARCH mice exhibited negative values, which are indicative of conditioned aversion. This point is further strengthened when one considers that these data are normalized to the pre-test condition and subjects were conditioned in a biased drug-assigned manner.

2. For the statistical data analyses, there are several concerns. First, it appears that the same data have been incorporated into the values for comparison twice. The preference scores for post-test were normalized to the pre-test, and then a t-test was performed on the pre-test values compared to the post-test/pre-test values. It would be more appropriate to either compare that actual values for all four conditions with an ANOVA (with correction for multiple comparisons) or conduct between group analyses on the normalized post-test data only. Further, it is unclear as to what the pre-test data on the graph actually represent - if these were 'Normalized Preference' to the pre-test condition (as the Y axis indicates), one would expect values of '0' (assuming subtraction was performed for normalization); this is not consistent with what is presented on the graph with the scatter dot data points and thus clarification is warranted.

3. The data suggest that activation of the IPN-LDTg pathway may mediate an aversive signal during real-time choice behavior, and that inhibition of this pathway can block the development of a conditioned environmental aversion to a high dose of nicotine. However, the data provided do not specifically demonstrate that nicotine aversion (e.g., aversion behavior to the intake of nicotine) is regulated by inhibition of the pathway. Thus, it is recommended that the title be modified since the terminology may be confusing to the reader. The abstract and conclusions should also more specifically indicate 'conditioned aversion to nicotine'.

4. For the CPP, it would be of benefit to indicate the time of day for the behavioral test (e.g., 5th day), since vehicle was administered in the morning and nicotine in the afternoon during conditioning sessions. If the behavioral test session varied between morning and afternoon among subjects, was this randomly assigned between groups?

5. For electrophysiological studies, the number of cells examined is indicated, but the number of subjects is not provided. Were these cell numbers from only one subject for each group? The number of animals per group, in addition to number of cells, need to be more clearly indicated.

6. Supplementary figure 2 demonstrates that activation of IPN terminals in the LDTg does not elicit aversion in a real-time preference paradigm. However, the authors do not specifically provide evidence that ChR2-expressing axons were localized in the LDTg in these mice. Given that a general ChR2 virus was injected into the IPN, is it possible that the subpopulation projecting to the LDTg was not infected in appropriate numbers to elicit a behavioral effect?

Minor

1. Please specify the age range of the mice examined for all studies.

2. Please specify the AAV serotype for all viral vectors.

Response to Reviewers' comments:

We thank the reviewers for their extremely helpful comments. We think that the changes suggested have made our manuscript much stronger, and we have responded to each comment in bold text below.

Reviewer #1

1. Supplementary figure 1 seems like a critical part of the manuscript and should be moved from the supplement to the main text (i.e. it should be figure 1) since the effect of direct, non-specific activation of the IPN by channelrhodopsin on mouse behavior has not been shown before.

We agree, and have moved Supplementary Figure 1 to the main text as the new Figure 1.

2. It would be helpful if images showing the projection from the IPN to the LDTg, taken from mice expressing ChR2 in the IPN, were included in the manuscript (or at least YFP expression in the LDTg after virus injection in the IPN).

We have now included images of the LDTg with labeled IPN terminals in Supplemental Figure 3D.

3. Figure 3C is difficult to understand because the groups are not labeled and a key for the colors is not included. Also, data for this experiment should be analyzed by two-way ANOVA with test day and virus expression as main factors.

We thank Reviewer #1 for bringing this to our attention and agree that the suggested changes to the analysis is an improved approach. We have added color keys to Figure 3C (now Figure 4C), and have revised our analysis of conditioned place aversion data. We have now used a 2-way RM ANOVA with test day and virus expression as main factors, as suggested, and used a Bonferroni post-hoc test for multiple comparisons (Figure 4C, Supplemental 9A-C).

4. It appears as if the manuscript was formatted for a short communication. In addition to point 1 above, the discussion should be expanded to include how the new data fits with what is known regarding the IPN based on recent literature (i.e., Ables et al., 2017 PNAS, Koppensteiner et al., 2017 Cell Reports; Molas et al., Nat Neurosci., 2017; etc.).

Yes, this manuscript was originally formatted as a short communication and we have now expanded the discussion to include the references cited here, as well as other relevant recent findings. Discussion of these references in particular can be found on page 10 lines 2-14.

Minor:

1. Supplementary figure 8 appears out of place and should be moved to the beginning of the supplementary data.

We have moved Supplementary Figure 8 to the beginning of the supplementary data (Supplementary Figure 1), as suggested.

Reviewer #2

1. Cell-population specificity is missing. ChR2 and ARCH3 were not delivered in a cell specific manner using Cre-mice or similar. Viral injections were done in the ventral/caudal area of the IPN and avoided the rostral part of the IPN (IPR) to prevent infection of the VTA. However the IPR contains a specific cell population that also projects to the LDTg that expresses $\alpha 5$ (Hsu et al. J.Neurosci 2013), $\alpha 4$ and somatostatin (Zhao-Shea et al, Curr Biology 2013) and NOS (Ables et al. PNAS 2017) that may have an opposite role in nicotine aversion (Ables 2017) Also the intermediate IPI seems to contain a specific cell population that when activated increases anxiety (Zhao-Shea Nat comm 2014). This lack of cell specificity should be discussed and maybe it should be mentioned in the abstract that mostly the ventral part of the IPN was targeted as it is possible that different populations in the IPN might not cause aversion to nicotine.

We have included these considerations in our expanded discussion section on page 9 lines 1-18.

2. The studies that identified $\alpha 2$ nAChR presynaptic receptors in the IPN-LDTg connection did not identify the alpha subunit. Could the authors discuss or test more blockers to test the possibility that $\alpha 5$, $\alpha 2$ or other alpha subunits could be involved in the response to nicotine.

We have included these considerations in our expanded discussion section on page 9 lines 23-26.

3. The authors pair optogenetic manipulations with nicotine at different doses (2 aversive doses and a lower dose found in serum levels of smokers). The lower concentration found in smokers does not affect ChR2 light-evoked IPSC amplitude or frequency. And ARCH was also paired with only a high aversive nicotine dose. Also the lower photostimulation of 1 Hz does not affect place aversion. How these high nicotine doses and photostimulation frequencies relate to physiological levels or levels found in smokers. Could the authors describe this in more detail.

We agree that this is an important point and have expanded the discussion of the relationship between stimulation frequency and nicotine dose on page 5 lines 19-22.

Reviewer #3

1. Overall, the behavioral data are not very convincing as significant overlap is found between groups, with outliers likely contributing to the statistical differences found with one-tailed t-tests. For instance, with regard to Figure 3, the authors conclude that 'EYFP controls developed significant aversion to the nicotine-paired side, while ARCH mice did not'. However, more than half of the ARCH mice exhibited negative values, which are indicative of conditioned aversion. This point is further strengthened when one considers that these data are normalized to the pre-test condition and subjects were conditioned in a biased drug-assigned manner.

While there is some overlap between groups, we tested for potential outliers in all experiments, as described in the Methods sub-section 'Data Analysis'. Statistically defined outliers were excluded from the presented data. We have also now analyzed the behavioral data using the more stringent 2-way RM-ANOVA as recommended by Reviewer 1. Presentation of the normalized data was used to most effectively communicate the key findings from these experiments. We have clarified that the statistical analyses were conducted on the raw preference scores, rather than on the normalized data. The raw data are now also shown in Supplementary Figure 9A.

Since some of the ARCH mice did indeed exhibit conditioned aversion to nicotine, we have re-phrased our conclusion to more precisely reflect our statistical findings (page 7, below line 10).

While there are concerns regarding interpretation of conditioned behavior using a biased apparatus, the apparatus in our experiment is not considered biased as indicated by no differences in side preference from control groups across several studies conducted in our laboratory. We do appreciate that there are still some concerns using a biased drug-assignment procedure in an unbiased apparatus as a shift in preference may be a result of regression toward the mean. However, this has been discussed as being an unlikely factor in observable shifts in preference or avoidance (Bozarth, 1987). Consistent with this discussion, no such shift in preference or avoidance in the EYFP+vehicle control group was observed in the current study. Taken together, we hope that these changes and clarifications have strengthened our interpretation and conclusions from the behavioral data.

2. For the statistical data analyses, there are several concerns. First, it appears that the same data have been incorporated into the values for comparison twice. The preference scores for post-test were normalized to the pre-test, and then a t-test was performed on the pre-test values compared to the post-test/pre-test values. It would be more appropriate to either compare that actual values for all four conditions with an ANOVA (with correction for multiple comparisons) or conduct between group analyses on the normalized post-test data only. Further, it is unclear as to what the pre-test data on the graph actually represent - if these were 'Normalized Preference' to the pre-test condition (as the Y axis indicates), one would expect values of '0' (assuming subtraction was performed for normalization); this is not consistent with what is presented on the graph with the scatter dot data points and thus clarification is warranted.

We thank the reviewer for the opportunity to clarify the description and presentation of the data. We apologize for the confusion and hope that we have clarified these points in the text. As mentioned above, we have altered our statistical analyses to the more appropriate 2-way RM ANOVA with Bonferroni post-hoc tests, and have clarified that the statistics were done using the raw preference scores. We have also described the normalization procedure and clarified the data presented in Figure 4C (page 7, starting on line 9).

3. The data suggest that activation of the IPN-LDTg pathway may mediate an aversive signal during real-time choice behavior, and that inhibition of this pathway can block the development of a conditioned environmental aversion to a high dose of nicotine. However, the data provided do not specifically demonstrate that nicotine aversion

(e.g., aversion behavior to the intake of nicotine) is regulated by inhibition of the pathway. Thus, it is recommended that the title be modified since the terminology may be confusing to the reader. The abstract and conclusions should also more specifically indicate 'conditioned aversion to nicotine'.

While we did not assess changes in ongoing nicotine intake, which is often used to assess nicotine aversion, nicotine aversion also commonly refers to the outcome measure in the conditioned place aversion paradigm (Frahm et al., 2011). To avoid confusion and address the reviewer's concerns, we have edited the abstract and other aspects of the text to more specifically state that we assessed nicotine aversion using a conditioned place aversion paradigm.

4. For the CPP, it would be of benefit to indicate the time of day for the behavioral test (e.g., 5th day), since vehicle was administered in the morning and nicotine in the afternoon during conditioning sessions. If the behavioral test session varied between morning and afternoon among subjects, was this randomly assigned between groups?

We thank the reviewer for bringing this oversight to our attention. We have clarified in the Methods sub-section 'Behavioral Testing' (sub-heading 'Conditioned Place Aversion') that pre- and post-tests were conducted at an intermediate time point, so as not to overlap with the time of day of either the AM or PM conditioning sessions. Additionally, this section now describes how animals were assigned to treatment groups.

5. For electrophysiological studies, the number of cells examined is indicated, but the number of subjects is not provided. Were these cell numbers from only one subject for each group? The number of animals per group, in addition to number of cells, need to be more clearly indicated.

We have now included the number of mice used in each experiment, in addition to the number of cells tested.

6. Supplementary figure 2 demonstrates that activation of IPN terminals in the LDTg does not elicit aversion in a real-time preference paradigm. However, the authors do not specifically provide evidence that ChR2-expressing axons were localized in the LDTg in these mice. Given that a general ChR2 virus was injected into the IPN, is it possible that the subpopulation projecting to the LDTg was not infected in appropriate numbers to elicit a behavioral effect?

We apologize for the confusion with this figure. We have now included the data from Supplemental Figure 2 into the main Figure 2 (formerly Figure 1). The data shown was for 1 Hz stimulation of LDTg terminals, which did not result in behavioral avoidance. We hope that moving this data to Figure 2 clarifies our interpretation, as it can be compared to the effects of 20Hz stimulation, which did result in behavioral avoidance. Additionally, the data for 20Hz stimulation and 1 Hz stimulation were from the same animals tested on different days (described in detail in Methods sub-section 'Behavioral Testing', sub-heading 'Real-time preference test'). Therefore, if the lack of behavioral effect from 1 Hz stimulation was due to lack of ChR2 expressing terminals in the LDTg, we would not have seen a behavioral effect of 20Hz stimulation in the same mice.

We have also included representative images of LDTg sections that include labeled IPN terminals (Supplemental Figure 3D).

Minor

1. Please specify the age range of the mice examined for all studies.

We have included this information in the Methods section, sub-section 'Animals'. We also repeat this information in the Methods section, sub-section 'Surgical Procedures and Viruses Used'.

2. Please specify the AAV serotype for all viral vectors.

We have included this information in the Methods section, sub-section 'Surgical Procedures and Viruses Used'.

REVIEWERS' COMMENTS:

Reviewer #1 (Remarks to the Author):

The reviewers have done an outstanding job in revising their manuscript to satisfy all reviewers' concerns. I only have one minor comment.

1. Lines 263-264, the specific paper that identifies the IPN's role in nicotine withdrawal-induced anxiety is Zhao-Shea et al., Nature Communications, 2015. Zhao-Shea et al., 2013 Current Biology is cited but this paper deals predominantly with physical withdrawal symptoms.

Reviewer #2 (Remarks to the Author):

The authors have responded to most of the comments of reviewers 2 and 3 and have made appropriate changes in the manuscript and figures. However there are three points that should still be addressed:

Reviewer #2 point 3:

There was not enough discussion on this point. Does a physiological nicotine dose in a smoker would affect conditioned place aversion and how this compares to the levels achieved by optogenetic manipulations.

Since:

Chr2 photostimulation: conditioned place aversion (Fig 1)

Nicotine high dose (in EYFP mice): conditioned place aversion (Fig 4)

Arch photoinhibition: almost aversive (Fig S9)

Arch photo inhibition + nicotine: no more aversive... (Fig 4)

How these results relate to smoking dependence.

Reviewer #3 point 1 and 2:

The new statistic analyses with 2 way RM ANOVA in figure 4 are correct however the unpaired t-test for figure 4C should be changed to a non parametric t-test like the Tukey's test.

Reviewer #3 point 3:

The abstract changes are appropriate but the title has not been modified to indicate 'conditioned aversion to nicotine', instead of nicotine aversion. The paper the authors refer in their reply (Frahm 2011) had both measurements of conditioned place aversion and nicotine intake.

P.S. We recommend that you upload the step-by-step protocols used in this manuscript to the Protocol Exchange. Protocol Exchange is an open online resource that allows researchers to share their detailed experimental know-how. All uploaded protocols are made freely available, assigned DOIs for ease of citation and fully searchable through nature.com. Protocols can be linked to any publications in which they are used and will be linked to from your article. You can also establish a dedicated page to collect all your lab Protocols. By uploading your Protocols to Protocol Exchange, you are enabling researchers to more readily reproduce or adapt the methodology you use, as well as increasing the visibility of your protocols and papers. Upload your Protocols at

www.nature.com/protocolexchange/. Further information can be found at www.nature.com/protocolexchange/about.

P.S. To help the scientific community achieve unambiguous attribution of all scholarly contributions, Nature Communications encourages all authors to create and link an ORCID identifier to their account. Please ensure that all co-authors are aware that they can add their ORCIDs to their accounts, so that it will display on this paper. If they so wish, they must do so before the paper is formally accepted. It will not be possible to add ORCIDs post-acceptance, e.g. at proof. To add an ORCID please follow these instructions:

1. From the home page of the [MTS](http://mts-ncomms.nature.com/cgi-bin/main.plex) click on 'Modify my Springer Nature account' under 'General tasks'.
2. In the 'Personal profile' tab, click on 'ORCID Create/link an Open Researcher Contributor ID (ORCID)'. This will re-direct you to the ORCID website.
- 3a. If you already have an ORCID account, enter your ORCID email and password and click on 'Authorize' to link your ORCID with your account on the MTS.
- 3b. If you don't yet have an ORCID account, you can easily create one by providing the required information and then clicking on 'Authorize'. This will link your newly created ORCID with your account on the MTS.

Response to Reviewers' comments:

Reviewer #1

The reviewers have done an outstanding job in revising their manuscript to satisfy all reviewers' concerns. I only have one minor comment.

1. Lines 263-264, the specific paper that identifies the IPN's role in nicotine withdrawal-induced anxiety is Zhao-Shea et al., Nature Communications, 2015. Zhao-Shea et al., 2013 Current Biology is cited but this paper deals predominantly with physical withdrawal symptoms.

We have replaced the 2013 reference with the more appropriate 2015 reference.

Reviewer #2

The authors have responded to most of the comments of reviewers 2 and 3 and have made appropriate changes in the manuscript and figures. However there are three points that should still be addressed:

Reviewer #2 point 3:

There was not enough discussion on this point. Does a physiological nicotine dose in a smoker would affect conditioned place aversion and how this compares to the levels achieved by optogenetic manipulations.

Since:

Chr2 photostimulation: conditioned place aversion (Fig 1)

Nicotine high dose (in EYFP mice): conditioned place aversion (Fig 4)

Arch photoinhibition: almost aversive (Fig S9)

Arch photo inhibition + nicotine: no more aversive... (Fig 4)

How these results relate to smoking dependence.

We have attempted to more clearly describe what we believe the relationship between optogenetic activation and different nicotine concentrations to be, but

are not entirely clear on what the reviewer is asking. The aim of our experiments was to elucidate the pathway that underlies the acutely aversive response to nicotine, an established predictor of future nicotine dependence. To do this, we used doses of nicotine that result in brain/plasma concentrations of nicotine that are physiologically relevant in humans (i.e. they can be achieved through tobacco smoking). We did not directly test whether alterations in this pathway change the rewarding and aversive effects of nicotine in mice dependent on nicotine (i.e. following chronic treatment). Our results show that we can recapitulate the aversive effects of nicotine by increasing stimulation through ChR2 in specific parts of this pathway (IPN cell bodies or IPN terminals in the LDTg). Further, if we block the nicotine-induced stimulation of GABA release from the IPN terminals in the LDTg with photoinhibition using ARCH, we block the aversive effects of nicotine. These results provide insight into mechanisms that could potentially be targeted to mitigate the development of nicotine dependence.

Reviewer #3 point 1 and 2:

The new statistic analyses with 2 way RM ANOVA in figure 4 are correct however the unpaired t-test for figure 4C should be changed to a non parametric t-test like the Tukey's test.

We are not completely certain what the reviewer is requesting, as changing the post-hoc analysis to a non-parametric test would require we used a non-parametric omnibus test. However, we agree that the 2-way RM ANOVA, a parametric test, is the most appropriate analysis. In Fig. 4C, we used a 2-way RM ANOVA with a Bonferroni post-hoc test for multiple comparisons. We ran the statistics on the raw data, as described in the manuscript (Supplemental Fig. 9A), and displayed the normalized data for presentation in the main figure. In light of this reviewer's comment, we have evaluated whether an unpaired t-test was the correct test for Fig 4D. We compared the change in preference after nicotine conditioning between two unmatched groups (ARCH and EYFP) using a two-tailed, unpaired t-test. Since our data fit the assumptions of the t-test and the Bonferroni correction is more stringent compared to Tukey's, we contend that our use of a parametric test for this analysis is appropriate.

Reviewer #3 point 3:

The abstract changes are appropriate but the title has not been modified to indicate 'conditioned aversion to nicotine', instead of nicotine aversion. The paper the authors refer in their reply (Frahm 2011) had both measurements of conditioned place aversion and nicotine intake.

'Nicotine aversion' is commonly used to refer to a variety of paradigms that assess the aversive effects of nicotine. There are many papers that include the word 'aversion' or 'aversive' in the title that exclusively relied on conditioned aversion assays. Some examples include: (Lammel et al., 2012) "Input-specific control of reward and aversion in the ventral tegmental area"; (Tan, Bishop, Lauzon, Sun, & Laviolette, 2009) "Chronic nicotine exposure switches the functional role of mesolimbic dopamine transmission in the processing of nicotine's rewarding and aversive effects."; (Danjo, Yoshimi, Funabiki, Yawata, & Nakanishi, 2014) "Aversive behavior induced by optogenetic inactivation of ventral tegmental area dopamine neurons is mediated by dopamine D2 receptors in the nucleus accumbens"; (Laviolette, Alexson, & van der Kooy, 2002)-" Lesions of the tegmental pedunculopontine nucleus block the rewarding effects and reveal the aversive effects of nicotine in the ventral tegmental area."

While we maintain that our title is appropriate, we will defer to the editors on this issue.